


# Characterization of transport from the Asian summer monsoon anticyclone into the UTLS via shedding of low-potential vorticity cutoffs

Jan Clemens[1,4], Felix Ploeger[1,2], Paul Konopka[1], Raphael Portmann[3], Michael Sprenger[3], and Heini Wernli[3]

[1]Institute for Energy and Climate Research: Stratosphere (IEK–7), Forschungszentrum Jülich, Jülich, Germany.
[2]Institute for Atmospheric and Environmental Research, University of Wuppertal, Wuppertal, Germany.
[3]Institute for Atmospheric and Climate Science, ETH Zürich, Zürich, Switzerland
[4]Jülich Supercomputing Centre, Forschungszentrum Jülich, Jülich, Germany

**Correspondence:** j.clemens@fz-juelich.de

**Abstract.** Air mass transport within the summertime Asian monsoon circulation provides a major source of anthropogenic pollution for the upper troposphere and lower stratosphere (UTLS). Here, we investigate the quasi-horizontal transport of air masses from the Asian summer monsoon anticyclone (ASMA) into the extratropical lower stratosphere and their chemical evolution. For that reason, we developed a method to identify and track the air masses exported from the monsoon. This
method is based on the anomalously low potential vorticity (PV) of these air masses (tropospheric low–PV cutoffs) compared to the lower-stratosphere, and uses trajectory calculations and chemical fields from the Chemical Lagrangian Model of the Stratosphere (CLaMS). The results show evidence for frequent summertime transport from the monsoon anticyclone to mid-latitudes over the North Pacific, even reaching high latitude regions of Siberia and Alaska. Most of the low–PV cutoffs related to air masses exported from the ASMA have lifetimes shorter than one week (about 90%) and sizes smaller than 1 percent
of the northern hemisphere (NH) area. The chemical composition of these air masses is characterised by carbon monoxide, ozone and water vapour mixing ratios at an intermediate range between values typical for the monsoon anticyclone and the lower-stratosphere. The chemical evolution during transport within these low–PV cutoffs shows a gradual change from characteristics of the monsoon anticyclone to characteristics of the lower stratospheric background during about one week, indicating continuous mixing with the background atmosphere.

## 1 Introduction

The Asian summer monsoon anticyclone (ASMA) is the dominant circulation pattern in the summertime upper troposphere and lower-stratosphere UTLS (Randel and Jensen, 2013). The anticyclonic circulation develops as a response to diabatic heating associated with convection over South Asia (Gill, 1980; Rodwell and Hoskins, 1995) and lasts from around June to August, but with strong year-to-year variability (Santee et al., 2017). Moreover, considerable sub-seasonal variability exists in the strength
of the anticyclone, being largely linked to variability in convection (Randel and Park, 2006). Transport of polluted air masses from the boundary layer through the ASMA circulation into the UTLS has a significant impact on the chemical composition



of the UTLS and even the deep stratosphere, as shown from satellite observations of hydrogen cyanide (HCN) (Randel et al., 2010) and baloon-borne measurements of water vapor, ozone and aerosols (Brunamonti et al., 2018). It is the combination of strong pollution sources in South-East Asia, intense convection over this region, and confinement in the anticyclonic UTLS

flow which makes this transport particularly efficient as a pathway for pollution into the UTLS.

From a large-scale perspective, the polluted air masses in the ASMA have been shown to take two main pathways: a fast one into the extratropical lowermost stratosphere and a slow one into the tropical lower-stratosphere, from where the air may further ascend deep into the stratosphere (Ploeger et al., 2017). In more detail, convective uplift reaches to about 370 K potential temperature (Tzella and Legras, 2011; Bergman et al., 2012), with contributions from very different source regions

(Tissier and Legras, 2016), and largely abates below the local tropopause (von Hobe et al., 2021). Subsequent upward transport across the monsoon tropopause is related to slow upwelling and positive diabatic heating rates of around 1K day$^{-1}$ (von Hobe et al., 2021), and is in the horizontal plane characterized by an anticyclonic, spiralling motion (Vogel et al., 2019). In this vertical range, anomalous trace gas distributions indicate confinement in the ASMA (Park et al., 2009), with tracer anomalies correlating well with low PV anomalies both in the time-mean climatology and with regard to day-to-day variability (Garny

and Randel, 2013). The fact that the PV field shows a clear minimum in the monsoon UTLS can be used to define the edge of the ASMA core as the maximum horizontal gradient of PV when going from inside of the ASMA to the outside (Ploeger et al., 2015).

However, the confinement in the monsoon anticyclone is not perfect and some horizontal exchange occurs between the ASMA and its surroundings (Garny and Randel, 2016; Legras and Bucci, 2020). Export of air from the ASMA frequently

occurs when smaller-scale eddies are shed from the main anticyclone, so-called eddy shedding events (Hsu and Plumb, 2000; Popovic and Plumb, 2001). Idealized shallow-water models indicate different dynamical regimes for the eddy shedding to occur (Amemiya and Sato, 2018; Rupp and Haynes, 2021), but the relation to observed monsoon flow characteristics is still unclear. Air masses exported from the ASMA are typically characterized by anomalously low PV with respect to their surrounding background. It has been suggested that the air mass transport from the ASMA into the middle and high latitude extratropical

stratosphere may be grouped into (i) direct transport related to streamers along the ASMA edge, and (ii) zonal export from the anticyclone into the upper troposphere and subsequent transport to the extratropical stratosphere by Rossby-wave breaking along the subtropical jet, even remote from the ASMA region (Vogel et al., 2014; Kunz et al., 2015).

The fact that the air masses exported from the ASMA are characterized by anomalously low PV values and that PV is to first order materially conserved on isentropic levels, offers an opportunity to identify these air masses and to investigate their

pathways into the stratosphere. In this sense, air masses with anomalously low PV in the lowermost stratosphere (low–PV cutoffs) indicate air transported from the troposphere into the stratosphere. In principle, this transport may be reversible if PV was perfectly conserved and the low-PV cutoff returned to the tropospheric reservoir a few days later. If, however, the low-PV cutoff slowly erodes due to diabatic processes that increase its PV value, then irreversible troposphere-to-stratosphere transport occurs, associated with mixing of the original ASMA air with the stratospheric environment. Equivalent approaches of iden-

tifying low-PV cutoffs have been used to study stratosphere-troposphere exchange (STE) across the extratropical tropopause (e.g. Wernli and Sprenger, 2007; Homeyer and Bowman, 2013). In many past studies, the term cutoff is related to stratospheric



air with high PV in the troposphere and we emphasize here that in the present paper "cutoff" refers to the opposite process, i.e. tropospheric air of low PV in the stratosphere. An algorithm for tracking these cutoffs was recently developed by Portmann et al. (2021). In this study, PV streamers and cutoffs are identified with reference to a chosen critical PV contour on an isen-

trope, representing the tropopause. Furthermore, Kunz et al. (2011) showed that the dynamical tropopause on an isentrope is even better characterized by the maximum PV gradient (with respect to equivalent latitude) than a fixed value. This concept was further applied for diagnosing STE by Kunz et al. (2015). Their study showed a particularly high frequency of PV streamers over the eastern North Pacific in summer, indicating strong STE likely related to the monsoon anticyclone.

Here, we extend the method of Kunz et al. (2015) and further investigate the pathways of air masses from the ASMA

into the extratropical lower-stratosphere, motivated by the two questions: (i) What are the main pathways of isentropic, quasi-horizontal air mass transport from the monsoon anticyclone into the extratropical UTLS? (ii) What is the chemical composition of the air masses exported from the ASMA and how does the composition evolve during transport to the extratropics? For that reason, we carry out complementary PV-cutoff detection calculations for different PV values corresponding to the ASMA edge, the annual mean dynamical PV-gradient-based extratropical tropopause, and an even larger PV value characterizing the

summertime PV-gradient-based tropopause.

As described in Sect. 2, the results of the detection calculation for the three different PV values describe different stages of the low–PV cutoff after detaching from the ASMA and on its way into the extratropical lowermost stratosphere. Furthermore, low–PV cutoffs originating in the ASMA are identified by a filtering method that effectively excludes cutoffs that are not related to the ASMA. The filtering is based on a novel cutoff tracking algorithm. Characteristics like the size, lifetime, and

spatial distribution of the cutoffs, as well as their composition in terms of CO, $O_3$ and $H_2O$ mixing ratios (based on model simulations), are further investigated.

In Sect. 2, we explain the methodology to identify and track low–PV cutoffs in the lower stratosphere. Section 3.1 presents the global distribution and seasonality of the cutoffs, Sect. 3.2 the distribution of ASMA–related cutoffs, and Sect. 3.3 their characterization (e.g., lifetime, size). Section 4 presents an analysis of the chemical composition in terms of CO, $O_3$ and

$H_2O$ mixing ratios. Finally, we conclude with a discussion of uncertainties of the methodology and potential implications for measurement campaigns in Sect. 5.

## 2  Data and methods

### 2.1  ERA-Interim reanalysis and CLaMS

For our study we use Lagrangian transport calculations that are driven by reanalysis data. The used Chemical Lagrangian

Model of the Stratosphere (CLaMS) is a full chemical-transport model with a 4th order Runge-Kutta scheme for advection and a parameterization of sub-grid scale atmospheric mixing processes based on deformations in the large-scale flow (see McKenna et al., 2002). CLaMS can be used for full-blown chemical transport model multi-annual simulations of the chemical composition of the UTLS (see Pommrich et al., 2014) and pure trajectory calculations. The model numerics are calculated in hybrid vertical coordinates which are orography-following at the ground and transform smoothly into potential temperature





above, such that throughout the stratosphere transport is calculated in a diabatic framework, with vertical velocity deduced from the reanalysis diabatic heating rate.

In this paper, we consistently use the ERA-Interim reanalysis from the European Centre for Medium-range Weather Forecasts (ECMWF) for chemical transport calculations covering the period 2008 to 2018, as well as for cutoff detection and cutoff tracking. The ERA-Interim reanalysis offers 6-hourly wind fields with a horizontal resolution of 79 km and 60 vertical levels

up to 10 Pa (Dee et al., 2011). For this study the reanalysis has been interpolated to the 380K isentrope.

## 2.2 General approach

We consider cutoffs as air masses with tropospheric origin that are characterized by anomalously low PV embedded in the lower stratosphere. In addition, we particularly focus on cutoffs that can be traced back to the ASMA. Such a cutoff can be described by a closed PV contour of a specific value encircling the low–PV air mass. The PV contour that defines the ASMA

is found to be around 4.0 PVU (Ploeger et al., 2015). The climatological annual-mean tropopause on the 380 K isentrope corresponds to the 6.5 PVU contour, and the climatological tropopause during JJA to 7.5 PVU (see Sect. 2.3 for details).

Figure 1 schematically illustrates the transport from the ASMA into the lower stratosphere and the role of low-PV cutoffs for this transport along two characteristic pathways. The first, indirect transport pathway from the ASMA to the stratosphere starts with the shedding of a large-scale eddy at the eastern flank of the ASMA (4 PVU cutoffs). After the shedding process,

the eddy moves eastward over the North Pacific and decays into smaller low-PV cutoffs, all of which is happening within the troposphere. Subsequently air masses of the decaying eddy can be transported across the tropopause into the stratosphere. The second, direct transport pathway frequently starts at the north-eastern side of the ASMA with a strong bulging of the ASMA and the development of a low-PV streamer. The bulging of the ASMA is visible as deformation of the 4.0 PVU contour, while the streamers are mostly visible as the deformation of the 6.5 PVU contour. As illustrated in the schematic, the streamers

may contain 4.0 PVU cutoffs that are transported poleward. In the next phase, the streamer detaches as a 6.5 PVU cutoff. Subsequently, mixing processes further increase the PV value in these air masses such that they are only detectable as 7.5 PVU cutoff, and finally entirely mix with the background causing TST. Hence, the full lifecycle of a low–PV cutoff involves the sequence of air masses enclosed by 4.0, 6.5 and 7.5 PVU when being transported from the ASMA to high latitudes. In the following, we will investigate cutoffs with respect to these three PV thresholds to gain information about the cutoff life cycle,

since detachment from the monsoon anticyclone until mixing with the lower stratospheric background.

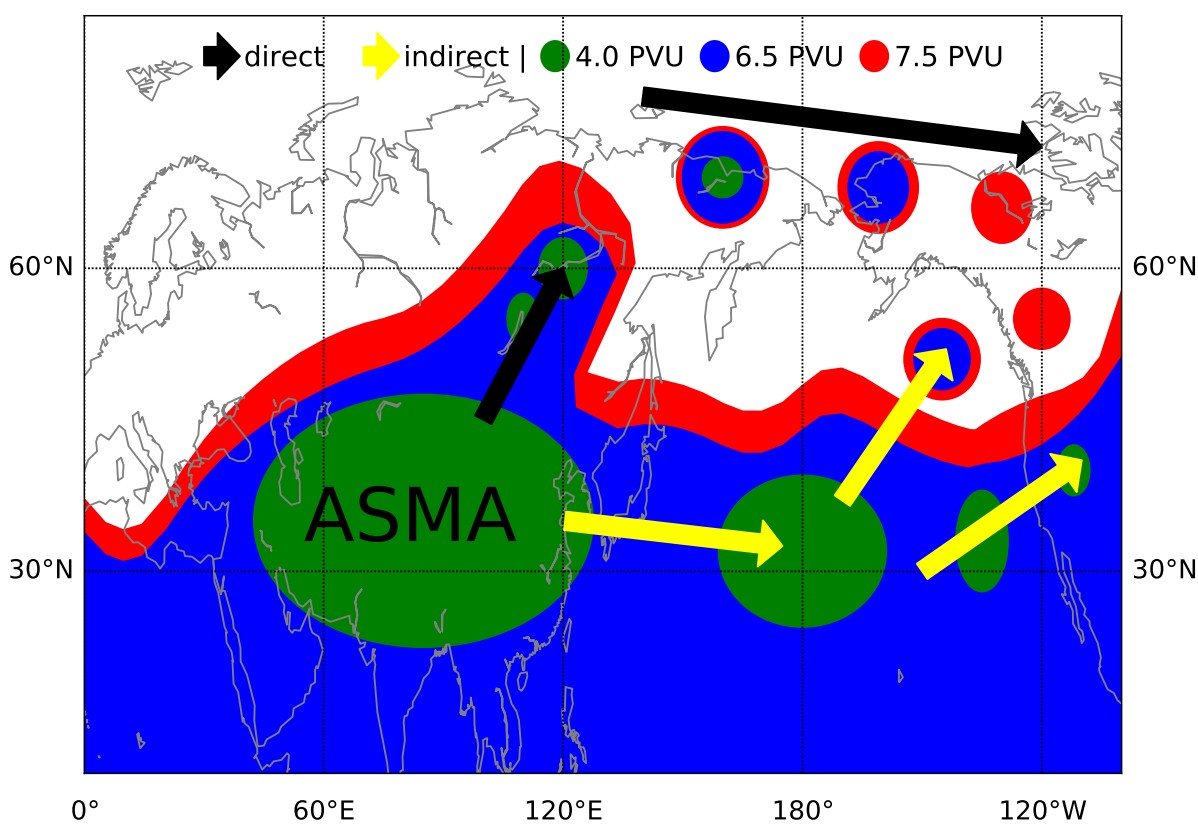

**Figure 1.** Schematic of the two idealized transport pathways from the ASMA into the stratosphere and the role of low-PV cutoffs on 380K for each of the pathways. Green areas mark cutoffs with PV≤4.0 PVU, blue marks areas with 4.0 PVU<PV<6.5 PVU. Red shows areas with 6.5 PVU <PV<7.5 PVU. The black arrows indicate the direct transport pathway and the yellow ones the indirect pathway.

## 2.3 Cutoff detection and tracking

Figure 2 presents the generation of cutoffs during the first week of July 2017 as an example of the direct pathway. On 1 July, the PV contours encircling the ASMA start to bulge at the north-eastern edge of the ASMA (around 45°N/100°E, see Fig. 2a). During the next three days, a low-PV streamer develops in that region that extends beyond 60°N, and covers parts of eastern Russia (Fig.2b-d). This streamer also contains small 4 PVU cutoffs (Fig. 2c,d). After 4 July, the streamer breaks up into several cutoffs. These cutoffs decay during the next three days (not shown). The magenta and yellow lines show the tracks of the low-PV anomalies, as determined from the 4.0 PVU (yellow) and the 6.5 PVU cutoff (magenta).






As examples for the indirect transport pathways, eastward eddy shedding or ASMA splitting events can be found when considering the 4.0 PVU contours. These cutoffs are shed at the eastern edge of the ASMA and propagate eastward within the
upper troposphere to Japan and the North Pacific (equatorward of the 6.5 PVU tropopause contour). Figure 3 shows the case of a cutoff that is shed from the ASMA (with respect to the 4 PVU contour) on 6 July 2017. After separating from the ASMA, the cutoff moves eastward over the North Pacific on the tropospheric side of the dynamical tropopause (7.5 PVU).

To extend the identification and the analysis of the cutoffs to climatological periods, we developed a dedicated algorithm. The general structure of the algorithm is illustrated in Fig. A1a. It consists of a part to determine the PV-gradient tropopause,
a part that detects the cutoffs, and a part that tracks the cutoffs. The cutoff identification and tracking in this paper is based on ERA-Interim reanalysis PV and wind data (see Sect. 2.1).

The tropopause detection algorithm is based on the work of Kunz et al. (2011). Therefore, the tropopause is defined as that PV contour where the gradient with respect to equivalent latitude of the product between zonal wind and PV has its maximum. These tropopause PV contours are used to determine cutoffs exported from the tropical troposphere. The tropopause detection
algorithm was used to identify the annual mean and the summer mean (JJA) tropopauses at the 380 K isentropic surface, which were found to be located along the 6.5 PVU and 7.5 PVU contours, respectively.

The low-PV cutoff detection algorithm was built upon ideas of Wernli and Sprenger (2007). It essentially consists of a flood-fill algorithm that is applied step-by-step to all points in the PV field. Those coherent areas in the field with a PV value lower than a given threshold are marked with a unique number. The output of the algorithm is a field, that marks grid points
within cutoffs with the corresponding cutoff number, and the remaining areas of the stratosphere and tropical troposphere with specific numbers. For the following analysis on the 380 K isentrope the thresholds were set to 4.0 PVU, 6.5 PVU and 7.5 PVU, representing the anticyclone edge, annual mean and summertime mean tropopauses respectively.

The tracking algorithm extends the methodology by Wernli and Sprenger (2007), and makes use of 2d Lagrangian trajectory calculations based on the Chemical Lagrangian Model of the Stratosphere CLaMS (McKenna et al., 2002). A similar algorithm
was recently developed by Portmann et al. (2021). The principal idea is to initialize air parcels within the detected cutoffs at time $t_0$, carry out a forward calculation of their trajectories over the time step $\Delta t$, and subsequently compare the new positions with the detected cutoffs at time $t_1 = t_0 + \Delta t$. The time step $\Delta t$ is here chosen as 6 hours, equal to the analysis time step in ERA-Interim.

The tracking calculation from time $t_0$ to time $t_1$ needs the two cutoff index fields at these times, which are created by the
detection algorithm. For each cutoff at time $t_0$ air parcels are initialized at the grid points within the cutoffs. These air parcels are calculated forward on an isentrope using the CLaMS trajectory module (based on 4th order Runge-Kutta scheme, see McKenna et al., 2002). For typical lifetimes of tropospheric cutoffs in the lower stratosphere of a few days, diabatic motions can be neglected and a 2D isentropic trajectory calculation for the tracking is a valid approximation. After the advection time step $\Delta t$, the new positions are compared with the cutoff field at time $t_1$. Four different cases for a given cutoff are possible.
First, the cutoff can persist, so that some of the trajectories end within the same cutoff one time step later. Second, the cutoff may vanish during the time step or reconnects to the troposphere. In that case, the forward trajectories will not match a cutoff at time $t_1$. The third possibility is the splitting of the cutoff into two or more cutoffs. A fourth and last possibility is the merging of





existing cutoffs into one new cutoff. In that case, the algorithm arbitrarily selects one of the previous cutoffs as the predecessor of the new cutoff. All these processes together can lead to complex connections between different cutoffs over multiple time steps, particularly if a large cutoff breaks down into many smaller ones. In the following, we will call a set of related cutoffs a "cutoff cascade".

Finally, the chemical composition of the cutoffs is studied using chemical fields ($H_2O$, $CO$, $O_3$) from simulations with the full-blown chemical transport model CLaMS. These simulations cover the period 2009–2018 and are driven with ERA–Interim reanalysis, such that the underlying meteorology is consistent with the detected PV cutoffs.

## 2.4 Filtering Asian monsoon cutoffs

This section presents the method to filter those cutoffs that transport air from the ASMA into the UTLS. This filtering method needs to take into account the existence of the two dominant pathways of low–PV cutoffs from the ASMA, as presented earlier. ASMA related cutoffs are distinguished from unrelated cutoffs with the help of lifetimes, sizes and locations of the cutoffs. The filter parameters were empirically chosen to fit results from visual inspection of the relevant cutoffs during the period July–September 2017, as further detailed below.

Figure 2 shows an example for the direct transport pathway from the ASMA into the lowermost stratosphere. During the event, several smaller 6.5 PVU cutoffs are found which propagate from Western Russia or Northern Europe into the ASMA region. Clearly, these cutoffs are not related to transport from the ASMA into the lowermost stratosphere. Such cutoffs unrelated to the ASMA often start westward of $45°E$ and have maximum sizes below 0.3% of the Northern hemisphere area during their lifetime. Hence, we use a westward longitude boundary of $45°E$ and a minimum size of 0.3% Northern hemisphere area to distinguish ASMA cutoffs from those unrelated to transport from the ASMA.

Figure 3 further presents an example for the indirect transport pathways, showing large eddies shedding eastward and remaining equatorwards of the tropopause during the first days after detachment from the ASMA. The detailed analysis of cutoffs in June and July 2017 shows sizes of these eddies between 0.5% and 3.3% of the NH. In contrast, the size of the observed ASMA varies between 5% and 10% of the NH during this time. Therefore, a critical size of 3.5% of the NH was chosen to distinguish the ASMA itself from these cutoffs during June and July. In late August and September the filtering by size does not work efficient, because of the decrease of the ASMA intensity and the frequent splitting of the 4.0 PVU contour. Eventually, the ASMA region inside the 4.0 PVU contour can be connected with the tropospheric reservoir. The ASMA then no longer appears as a 4.0 PVU cutoff and no filtering is needed. To further constrain the filters, the initialization of the tracking calculation was restricted to the region between $25°E$ and $180°E$, and between $15°N$ and $90°N$ where the ASMA is located, and the cutoffs of interest originate inside this region.

In summary, the following filter criteria are used to distinguish the cutoffs that are related to the ASMA from other cutoffs (see also Table 1). To filter cutoffs that propagate from far upstream into the ASMA region, we removed all cutoffs whose tracks start at longitudes westward of $45°E$. This filter was used for 6.5 PVU and 7.5 PVU cutoffs. It is not used for 4.0 PVU cutoffs, because they hardly originate so far upstream. To avoid small cutoffs that are not related to the ASMA, a minimum size of 0.3% of the NH area during cutoff lifetime has been chosen, as well as a minimum lifetime of 1 day.



**Table 1.** Used filter criteria for the cutoffs in the ASMA region, based on the case study.

|  | 4.0 PVU | 6.5 PVU | 7.5 PVU |
|---|---|---|---|
| minimum longitude of first tracking | - | 45° | 45° |
| minimal maximum size of a cascade during lifetime | 0.3%NH | 0.3%NH | 0.3%NH |
| minimum lifetime | 1 day | 1 day | 1 day |
| maximum cutoff size | 3.5 %NH | 3.5 %NH | 3.5 %NH |

Although the presented case study already underpins the chosen filters, we additionally validated the filtering method with the help of additional trajectory calculations. For this purpose, backward trajectories have been initialised inside all detected ASMA cutoffs in summer (JJAS) 2017 to trace the cutoff air masses backwards in time and investigate whether they indeed originated in the ASMA. As a result, 93% of the filtered 4.0 PVU cutoffs, 79% of the 6.5 PVU cutoffs and 96% of the 7.5 PVU cutoffs were identified as being of ASMA origin also in the backward trajectory calculation, further corroborating the filtering approach. Further details of the validation are discussed in the appendix C.







**Figure 2.** Example of the direct transport pathway from the ASMA into the lower stratosphere by cutoffs at 380K in early July 2017 (see dates on top of panels). The track of a small 4.0 PVU cutoff is shown in yellow and the track of the subsequent 6.5 PVU cutoff in magenta. The blue area marks the tropical troposphere (PV ≤ 6.5 PVU) and the green area marks the ASMA and the equatorial region (PVU ≤ 4.0 PVU). Red areas mark the mean tropical tropopause in JJA (PVU ≤ 7.5 PVU). Red areas mark the mean tropical tropopause in JJA (PVU ≤ 7.5 PVU). Black lines show tracks of the 6.5 PVU cutoffs and the text shows the maximum size of the cutoff in percent of the Northern Hemisphere.





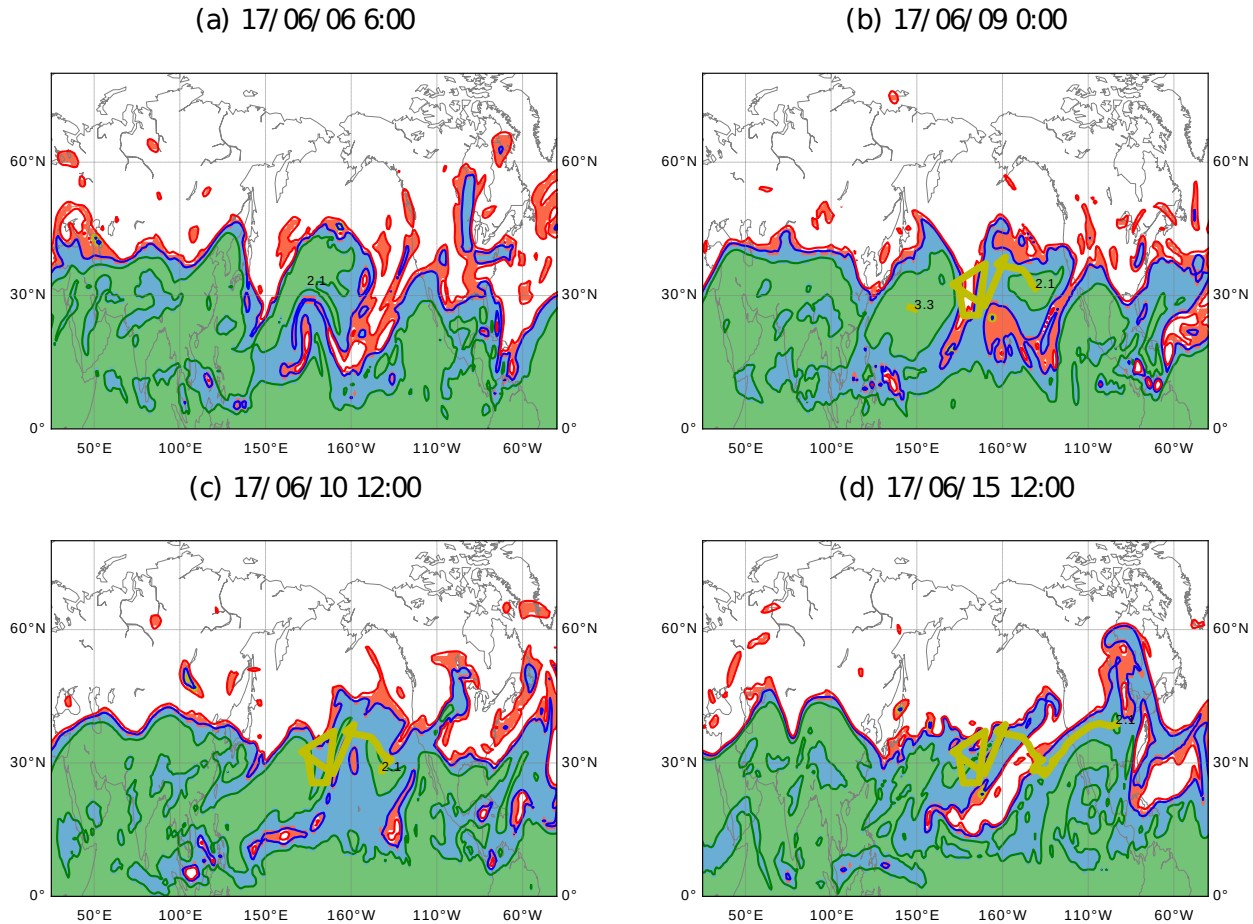

**Figure 3.** Example of the indirect transport pathway from the ASMA into the lower-stratosphere by two eastward shed cutoffs from the ASMA at 380K. Colors are as in Fig. 2. Yellow lines show tracks of the 4.0 PVU cutoffs. The largest cutoff (3.26% of NH) remains only a very short time over the West Pacific. The smaller cutoff (2.12% of NH) has a very long lifetime of about one week and travels to North America.

## 3 Diagnosis of troposphere-to-stratosphere transport associated with low-PV cutoffs

### 3.1 Seasonality in the extratropical UTLS

To interpret transport by low-PV cutoffs from the ASMA within a global context, we first consider the global frequency distributions of low-PV cutoffs (PV < 6.5 PVU) on 380 K in all seasons (Fig. 4). Following Wernli and Sprenger (2007), the presented seasonal mean frequency quantifies the fraction of times (relative to all time steps in a season) when a grid point is covered by a cutoff. For example, a value of 1% indicates that during 1% of the time this point is covered by a cutoff.





From autumn to spring (SON, DJF, MAM), similar patterns can be observed which are characterized by a band of high
cutoff activity that extends from the west coast of North America over the Atlantic and Europe to Central Asia. Transport
activity, as measured here by the cutoff frequency, peaks in the Atlantic-European region and is strongest during autumn
(SON). Furthermore, during autumn, winter and spring only weak transport activity related to low-PV cutoffs can be found
over the Pacific. These patterns agree well with results from past studies by Kunz et al. (2015).

During boreal summer the cutoff frequency distribution changes significantly. A strong peak in the low-PV cutoff frequency
emerges over the central North Pacific at about 30-50°N. The location of that peak downstream of the ASMA region and the
weak gradients in the PV distribution between the peak and South-East Asia suggest that these cutoffs are likely related to
transport from the Asian monsoon. This Pacific region with high cutoff frequency extends northward to Siberia and Alaska
indicating long-range transport from low to high latitudes. Hence, the Asian monsoon appears to strongly affect the distribution
of low-PV cutoffs in the NH and related transport, with air masses exported from the ASMA preferentially affecting the
stratosphere over the North Pacific.



(a) DJF

(b) MAM

(c) JJA

(d) SON

**Figure 4.** Global climatological frequency distribution of 6.5 PVU cutoffs in the four seasons. A frequency of 1% percent mean frequency means that during a season in 1% of the time-steps, a cutoffs covers the point.

## 3.2 Asian monsoon related transport

In the following, we focus on low-PV cutoffs on the 380 K isentrope that are directly related to the ASMA. Section 2 explained how these cutoffs were identified and that it is meaningful to identify low-PV cutoffs related to the ASMA with PV threshold values of 4.0, 6.5, and 7.5 PVU, respectively.

Figure 5a shows the JJAS distribution of ASMA cutoffs on 380K with respect to the 4.0 PVU contour. The frequency distribution shows the transport of air masses exported from the anticyclone. These ASMA–cutoffs mainly leave the anticyclone





above east China and then move downstream of the ASMA along about 35°N latitude, reaching Japan and the Pacific region. Some of the cutoffs propagate even to North America. The peak in monsoon air over Japan in this season has recently been found in independent analysis and has been described as a particular mode of the anticyclone (Honomichl and Pan, 2020). The

westward transport from the ASMA is comparably low, but weak cutoff activity extends zonally even to Morocco. Overall, the ASMA cutoffs mainly distribute between the 6.5 PVU tropopause on 380 K and the 4.0 PVU contour near the equator, showing that the related transport is mainly zonal and restricted to the troposphere. These features are in good agreement with the transport pathways from the ASMA related to eddy shedding found in theoretical studies (e.g. Hsu and Plumb, 2000).

The frequency distribution of cutoffs with PV < 6.5 PVU highlights the transport of air from the ASMA across the subtropics

into middle latitudes (Fig. 5b). As already indicated by the seasonal plots in Fig. 4, the cutoffs are most frequent above Siberia and the North Pacific. The location of peak intensity just downstream of the maximum frequency of ASMA–cutoffs indicates the relation between the two types of detected cutoffs, with 6.5 PVU cutoffs likely representing a later stage and the 4.0 PVU cutoffs representing an earlier stage of the cutoff life cycle.

Finally, the cutoff distribution with PV < 7.5 PVU shows the transport of ASMA air further into NH middle and high

latitudes (Fig. 5c). Peak intensity occurs above the North Pacific just north of the frequency peak of the 6.5 PVU cutoffs. This indicates that the 7.5 PVU cutoffs represent again a later transport stage when the ASMA air moves to even higher latitudes with continuously increasing PV, as compared to the 6.5 PVU cutoffs. In addition to the peak over the North Pacific, enhanced cutoff frequency also occurs above Siberia, and a few 7.5 PVU cutoffs even reach the pole. The 7.5 PVU cutoff frequency is higher than the frequency of 6.5 PVU cutoffs due to the larger extend of the former cutoffs.

Furthermore, the frequency distribution of ASMA cutoffs shows considerable sub-seasonal variability. ASMA cutoff activity starts in June and becomes strongest during July and August (see appendix Fig. B1). Also September still shows frequent ASMA cutoffs, with highest occurrence above Alaska.





(a) 4.0 PVU

(b) 6.5 PVU

(c) 7.5 PVU

**Figure 5.** Climatological frequency of ASMA-related cutoffs during JJAS, for (a) 4.0 PVU cutoffs, (b) 6.5 PVU cutoffs and (c) 7.5 PVU cutoffs. The blue lines indicate particular PV contours. A subset of the identified tracks with a lifetime of at least 4 days is plotted with black lines, where green dots symbolize the starting points and red dots the end points of the tracks. Here, 1% means that a point is covered with a cutoff in 1% of the time steps in JJAS.

## 3.3 Characterization of cutoffs in time and space

Cutoff characteristics like their lifetime, size and the maximum latitude reached, are closely related to transport and atmospheric composition. Figure 6 presents probability density functions (PDFs) and cumulative density functions (CDFs) of these





properties for the cutoffs related to the ASMA, with individual distributions shown for the three cutoff stages (4.0, 6.5 and 7.5 PVU), respectively.

The frequency distribution of lifetimes is shown in Fig. 6a (note the logarithmic scale). The distributions are similar for the three stages and characterised by a high frequency of short lifetimes and a low frequency of long lifetimes. Between 90% and 98% of the cutoffs exist for less than one week. Most of the cutoffs have a very short lifetime between 1-3 days and the distribution shows a long tail with rare events with lifetimes up to two weeks. The tail of the distribution is somewhat stronger for the cutoffs at later stages (6.5 and 7.5 PVU), indicating more long-lived events for these categories.

A frequency distribution of the maximum sizes in a cutoff cascade is shown in Fig. 6b,d (note the logarithmic scale). More than 50% of the cutoffs have a size between 0.3% and 0.75% of the NH area. The distributions have a strong tail up to sizes of about 2.0% (7.5 PVU), 2.7% (6.5 PVU) and 3.5% (4.0 PVU) of the NH area, and they all peak at the smallest size bin considered. Larger cutoffs appear to contribute more significantly to the ASMA cutoffs directly after detachment from the anticyclone (4.0 PVU cutoffs), compared to the cutoffs detected at later stages (6.5 and 7.5 PVU). Hence, the ASMA is a source of large-scale cutoffs, likely originating from shedding processes or even splittings of the ASMA. Splitting events happen mainly in August and September when the ASMA weakens and presumably explain most of the largest cutoffs.

A view on the distributions of the maximum latitude a cutoff reaches (Fig. 6c,f), shows that the ASMA cutoffs are mainly restricted to latitudes below 50°N, stressing the fact that they contribute mainly to zonal transport in the troposphere. On the other hand, the frequency of maximum latitudes for cutoffs at later stage (tropospheric 6.5 PVU cutoffs) peaks near 50°N. This again emphasizes the importance of considering the later stage of cutoff events for long-range meridional transport.

As the low-PV cutoffs decay, i.e. their PV increases to typical stratospheric background values, they become smaller and mix with stratospheric air. Figure 7 shows the evolution of the size of the tropospheric–6.5 PVU cutoffs with a lifetime longer than 5 days (note the logarithmic scale). In contrast to the size of the largest cutoff within a cascade (as in Fig. 6b,e) here the individual sizes of all cutoffs within a cascade are shown as a function of life time. Hence, intermediate sizes below the filter criteria of 0.3% are possible, as long as the size of one cutoff in the full cascade at some point exceeds the threshold. On average, the cutoff size decreases gradually over time, showing an approximately exponential decay with a half-life time of around 3 days. For some cutoffs, their size stays nearly constant for a while and then suddenly decreases, or even increases. This behaviour is in accordance with the variability of the cutoff pathways and with the physical processes that affect PV and therefore the size of the cutoff along these pathways. A sudden decay, for example, can appear if a cutoff crosses the tropopause, while a strong interim increase in size can result from the collision and merging of two cutoffs.





(a) Lifetimes  (b) Sizes  (c) Maximum latitude

(d) Lifetimes  (e) Sizes  (f) Maximum latitude

**Figure 6.** PDFs (top row) and CDFs (bottom row) for the lifetime, size and maximum latitude of ASMA-related cutoffs at 380K between 2008 and 2018. The frequencies are determined from the number of cutoffs.

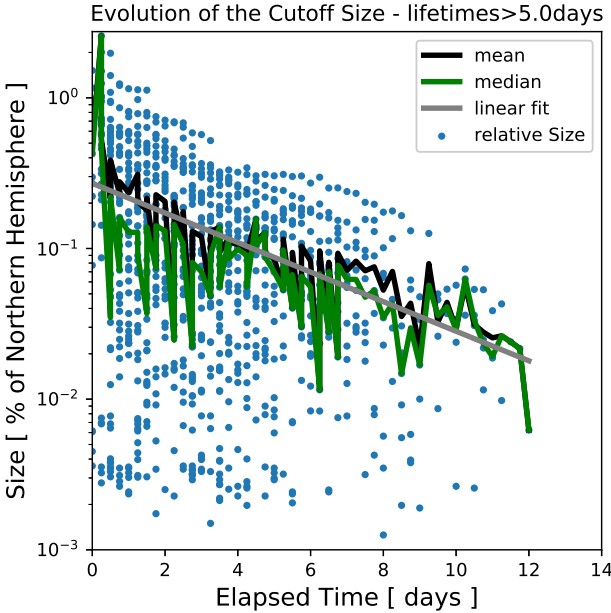

**Figure 7.** Decay of cascades of 6.5 PVU cutoffs, for cascades that life longer than 5 days. The blue dots show relative sizes in percent of the NH of each cutoff within the cascade. Note that a cutoff of a cascade can have sizes below 0.3% of the NH, while the largest cutoff of the cascade can still exceed the threshold of 0.3% at some time.

## 4 Chemical composition

The low-PV cutoffs separated from the Asian monsoon anticyclone are characterized by anomalous tracer concentrations indicating young tropospheric and highly polluted air masses. Hence, such cutoffs provide a pathway for polluted tropospheric air into the lowermost stratosphere. To further investigate the chemical evolution within the cutoff air we analyse the mixing ratios for CO, $H_2O$ and $O_3$ in the cutoffs, and compare them with typical values for the entire troposphere and stratosphere, respectively.

Figure 8 shows the mixing ratio distributions for CO, $H_2O$ and $O_3$. First, the calculated dynamical tropopause appears to be in good agreement with the chemical separation between the stratosphere and the troposphere, as mixing ratios for air masses characterized by PV values above and below the tropopause value clearly differ. As indicated in Fig. 8, the mixing ratio distributions for CO and $O_3$ for the entire atmosphere show a clear bimodal structure. The peak with high CO and low $O_3$ mixing ratios is related to the upper troposphere while the low CO and high $O_3$ peak is related to the lower stratosphere.

For $H_2O$, the separation between upper tropospheric and lower stratospheric mixing ratios in the global distribution appears less clear due to a large overlap between the two peaks. This overlap is likely related to the fact that for $H_2O$ the chemical separation is better described by the cold point tropopause than the dynamical tropopause used here.




The mixing ratios of air masses just exported from the ASMA are represented by the distributions for the 4.0 PVU cutoffs. These distributions show evidence that the ASMA contains the highest CO, the lowest ozone and highest water vapor mixing
ratios, corroborating the role of the ASMA as a source for tropospheric, highly polluted air.

The chemical evolution in the cutoff air masses becomes clear from comparison of the ASMA–cutoffs at different stages during their lifecycle (4.0, 6.5, 7.5 PVU cutoffs). At the later stage (6.5, 7.5 PVU), the cutoffs show relatively broad mixing ratio distributions that maximize between the tropospheric and the stratospheric peaks. These intermediate chemical characteristics of the cutoffs between troposphere and stratosphere are consistent with the evolution of the cutoffs during transport from the
ASMA into the lower stratosphere and related mixing with the surroundings. At the latest stage during the cutoff life cycle (7.5 PVU), just before mixing with the lower stratospheric background, the cutoffs are characterized by mixing ratios with the strongest stratospheric character. For these cutoffs the CO mixing ratios are lower and $O_3$ mixing ratios are higher compared to the 6.5 PVU cutoffs, and already close to the background values in the lower stratosphere.

### (a) CO            (b) H$_2$O            (c) O$_3$

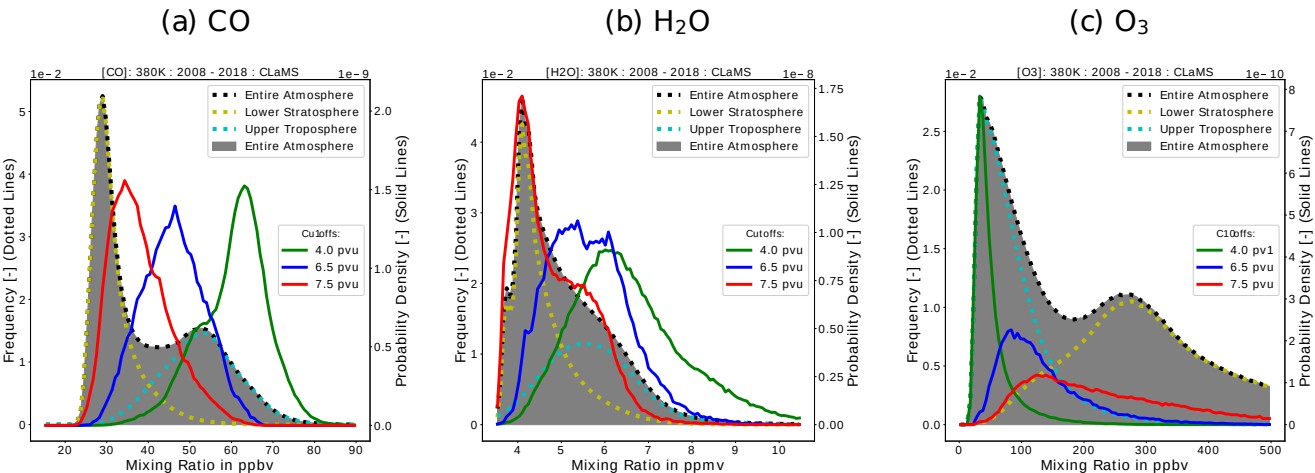

**Figure 8.** Chemical composition of the ASMA and tropospheric cutoffs at 380 K. Shown are histograms and PDFs of the mixing ratio of (a) CO, (b) H$_2$O and (c) O$_3$, all simulated with CLaMS. The histograms are plotted with dotted lines. The related axis is always on the left. The histograms show the frequency of mixing ratios within the different spheres of the atmosphere. Additionally, PDFs are shown with solid lines. The related axis is on the right. The probability density shows the frequency of mixing rations during the different stages of the cutoffs.

In the following, we further analyse the chemical evolution in the cutoffs over the cutoff life cycle.
To illustrate the early stage of the cutoff life cycle, Fig. 3 shows a cutoff that is shed from the ASMA (with respect to the 4.0 PVU contour) on 6 July 2017. After separating from the ASMA, the cutoff moves eastward over the North Pacific on the tropospheric side of the dynamical tropopause (7.5 PVU). The chemical composition in the cutoff changes continuously during the eastward propagation, without large steps (Fig. 9). CO shows a strong decrease in mixing ratio, caused by mixing with stratospheric background air and chemical loss, while $O_3$ mixing ratios are increasing during eastward propagation. The water
mixing ratio shows only a small positive trend.





The final stage of the cutoff life cycle before mixing with the stratospheric background is illustrated by showing an example of a 7.5 PVU cutoff in Fig. 10. The small 7.5 PVU cutoff forms around the 23 July 2017 above the North Pacific and moves further north-eastward until the 27 July. This cutoff moves fast over a long distance and transports tropospheric air deep into the high-latitude stratosphere. Also, the cutoff contains well-defined 6.5 PVU cutoffs (closed contours) for some time. After

passing over Alaska and parts of Canada the cutoff finally mixes with stratospheric air and disappears. As shown in Fig. 11, the CO and $H_2O$ mixing ratios slightly decrease over the first about 5 days while the $O_3$ mixing ratios weakly increase. After the 5th day stronger mixing ratio changes occur, likely related to interaction with other cutoffs and mixing over the Eastern part of North America.

A statistical analysis of the chemical evolution in all cutoffs during 2008 to 2018 is presented in Fig. 12. The figure shows

the mean evolution of CO, $H_2O$ and $O_3$ mixing ratios in the 4.0, 6.5 and 7.5 PVU cutoffs over one week (cutoffs with lifetimes of at least 3 days are included). For CO, the mean mixing ratio in ASMA cutoffs is about 60 ppbv. Over one week, the mean CO mixing ratio decreases by about 10% (Fig. 12a–c). A similar percentage change occurs for the 6.5 and 7.5 PVU cutoffs over the same period. This similar mean decrease rate for the different cutoffs is related to gradual mixing of the cutoffs with stratospheric air masses and chemical decay (the chemical lifetime of CO in the UTLS is about 2–3 months).

For $H_2O$, the mean mixing ratio in the 4.0 PVU cutoffs stays largely constant, related to the fact that $H_2O$ is controlled by processes mainly at the tropopause. Indeed, for the 6.5 and 7.5 PVU cutoffs, which represent cross-tropopause transport, also the $H_2O$ mean mixing ratio changes. However, this mean $H_2O$ change is very weak (about 10% over 1 week) compared to $H_2O$ changes at the tropical tropopause, related to the fact that the extratropical stratospheric $H_2O$ distribution is only very weakly affected by subtropical tropopause temperatures (Hoor et al., 2010), such that mixing with background air controls the

composition of the cutoffs. The stratospheric tracer $O_3$ shows increasing mixing ratios in the cutoff air masses over the cutoff life cycle (Fig. 12g–i). The strongest $O_3$ increase occurs in the 7.5 PVU cutoffs, when mixing with the high-ozone stratospheric background air occurs. This mixing changes the mean mixing ratio from about 250 to 375 ppbv, hence by about 50%.

The mixing of the cutoff with the stratospheric background and the related change in chemical composition can occur either gradually over the cutoff life cycle, or in strong individual events (e.g., sudden break-up of the cutoff). To further investigate the

respective roles of these two possibilities, Fig. 13 shows the PDFs of the mixing ratio changes (daily change relative to the net change over the entire lifetime) for CO, $H_2O$ and $O_3$ and for the 4.0, 6.5 and 7.5 PVU cutoffs. For all species, the distributions peak close to zero. For CO and $H_2O$, the peak occurs at weakly negative tendencies, consistent with the general decrease of mixing ratios over the life cycle. For $O_3$, on the other hand, the peak occurs at slightly positive mixing ratios, consistent with a general mixing ratio increase caused by mixing with stratospheric background air. The existence of these strong peaks at

very small changes shows the dominant role of slow and gradual processing in the cutoffs, related to continuous mixing with stratospheric background air, and additional effects of chemistry (decay for CO, production for $O_3$). However, strong singular mixing events are not entirely negligible for the chemical composition change in the cutoffs, as the apparent tails of the PDFs show. The non-zero PDF values at ±1 show a non-vanishing probability for the entire mixing ratio change in a cutoff to occur within one day, and can be related to a sudden cutoff decay or the merging of different cutoffs.





(a) CO                    (b) H$_2$O                    (c) O$_3$

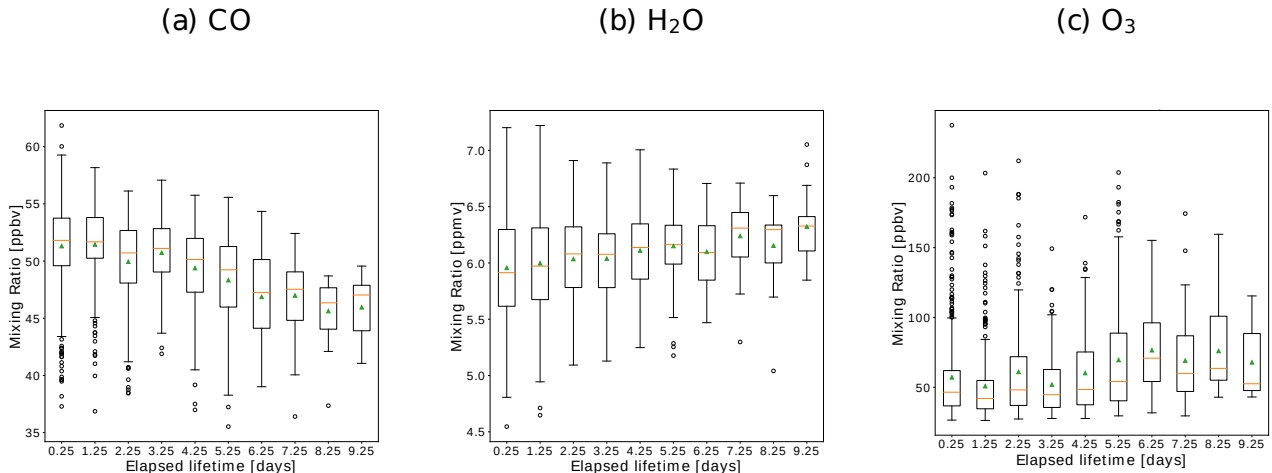

**Figure 9.** Chemical evolution of the chemical species (a) CO, (b )H$_2$O and (c) O$_3$ within the exemplary PV cutoff of Fig. 3. The box and whisker plots show the upper and lower quantile, the median (orange line) and the average (green pyramid) of the mixing ratios for each time step during the lifetime of the PV cutoff cascade. The black dots show outliers.





(a) 17/07/23 18:00

(b) 17/07/24 18:00

(c) 17/07/25 6:00

(d) 17/07/27 12:00

**Figure 10.** Example of an eastward shed 7.5 PVU cutoff from the ASMA at 380K. Colors are as in Fig. 2 The violet line shows the track of the moving 7.5 PVU cutoff.





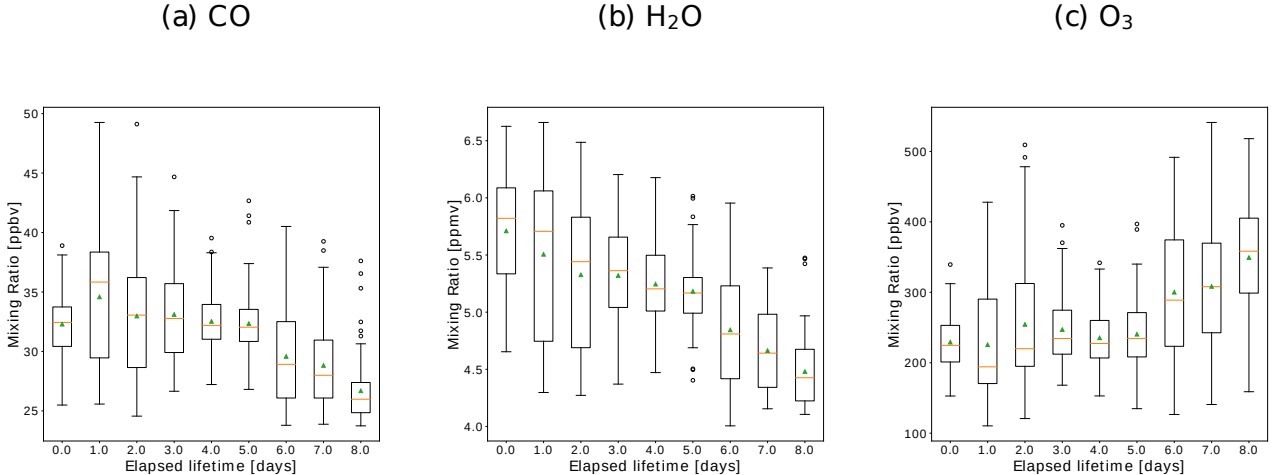

**Figure 11.** Chemical evolution of the chemical species (a) CO, (b) $H_2O$ and (c) $O_3$ along the exemplary PV cutoff shown in Fig. 10. Box and whisker plots are as in Fig. 9.





**Figure 12.** Mean evolution of CO (top row), $H_2O$ (middle) and $O_3$ (bottom) mixing ratios with respect to the 4.0 PVU cutoffs (left column), 6.5 PVU cutoffs (middle), and 7.5 PVU cutoffs (right). The mean evolution has been calculated by averaging over all cutoffs with a minimum lifetime of three days from 2008 to 2018. The red line shows the mean value and the green line the median for every time-step. Dark gray areas show the range between the upper and lower quantile and the light gray areas show the minimum-maximum range. The blue solid lines indicates the average at day 0.5 and the dotted blue lines the +/-10% range from this average.





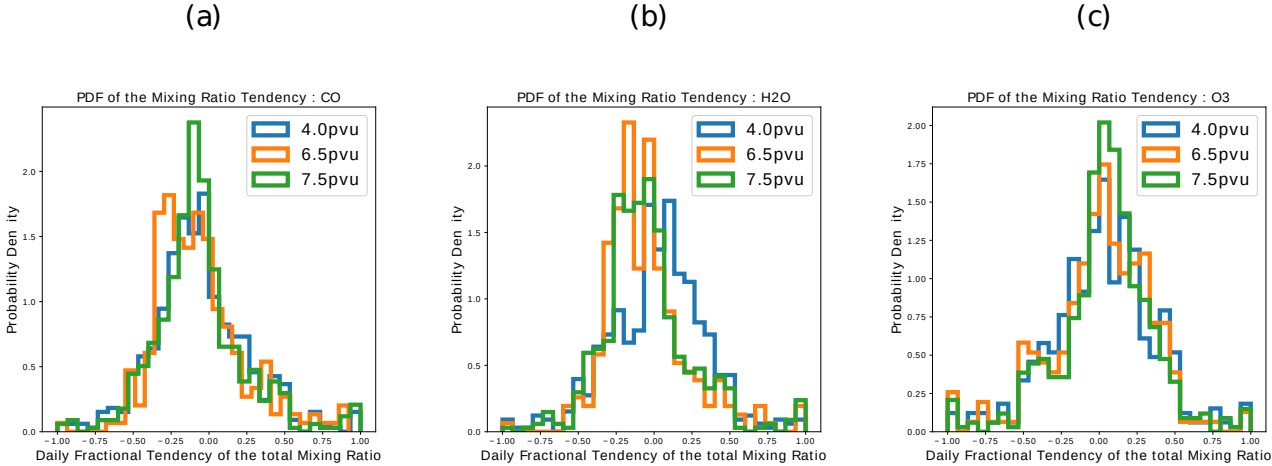

**Figure 13.** PDFs of the daily tendencies of the mixing ratios averaged within ASMA-related PV cutoffs. The tendencies - changes of mixing ratio per day - were normalized to the minimum-maximum mixing ratio range of each each cascade. Only cascades with lifetimes of 3 days or longer are included.

## 5 Discussion and conclusions

In this study, we investigated low-PV cutoffs that transport air masses from the ASMA into the extratropical lower-stratosphere. A new cutoff detection and tracking method was developed to identify the cutoffs in the Northern hemisphere PV field and to calculate their tracks. Spatial and temporal filters have been applied to relate the cutoffs to the ASMA.

Two dominant pathways for transport from the ASMA to the stratosphere have been identified which we termed the direct and indirect pathway, respectively. In both cases, the transport starts with cutoffs that shed as 4.0 PVU contours from the ASMA. These cutoffs show distinct high–CO, low–$O_3$ and high–$H_2O$ anomalies in the upper-troposphere.

In case of the indirect pathway and in the early stage of the cutoff life-cycle, the cutoffs are mainly located near the ASMA over Japan. These cutoffs are relatively large and propagate eastward. During eastward propagation the cutoffs decay and break up into smaller cutoffs. The remaining smaller cutoffs are transported further towards North America, and even further. Subsequently, the cutoffs decrease in size and their chemical characteristics change gradually towards lower CO and higher $O_3$ mixing ratios, likely related to continuous mixing with stratospheric background air that is not of ASMA origin.

In case of the direct pathway and in the early stage of the life-cycle, the cutoffs emerge from streamers at the north-eastern flank of the ASMA. These air masses can be identified as 4.0 PVU cutoffs within 6.5 PVU streamers and are of small to medium size. A few days later, the 6.5 PVU streamers break up and the PV of the cutoffs increases. Consequently, the remaining PV anomalies can only be identified with the criterion $PV \leq 6.5$ PVU (not 4.0 PVU anymore). In this stage, the cutoffs are located typically over East Russia or the North Pacific, still containing polluted air of ASMA origin. Again later, the PV of the 6.5 PVU cutoffs increases further, such that the cutoffs at even later stage can only be identified as anomalies with $PV \leq 7.5$ PVU, i.e., they form tropospheric cutoffs embedded in the lower stratosphere (recall that 7.5 PVU is the dynamical tropopause on 380





K). The transition from 6.5 PVU cutoffs to 7.5 PVU cutoffs takes place during the eastward transport above the North Pacific.

At the same time, the chemical composition of the cutoffs approaches stratospheric background values, with decreasing water vapor and CO and increasing $O_3$.

All types of cutoffs show skewed PDFs for their lifetimes and sizes, revealing the existence of a multitude of small-scale, short-lived cutoffs and only a few larger-scale and long-lived cutoffs. The temporal characterisation of the cutoffs reveals an average lifetime of the cutoffs of around 3 days. Moreover, more than 90% of the cutoffs have a lifetime below one week. Larger

cutoffs are more frequent during the early stage than during later stages. The cutoff size decreases approximately exponentially with a half-life of around 3 days. However, there is large case-to-case variability.

We found that cutoffs contribute to the transport from the ASMA to the stratosphere and hence to the troposphere-to-stratosophere mass flux. This leads to a couple of new questions in regard to their role for TST: (i) How large is the irreversible mass flux in relation to other irreversible TST processes? (ii) What types of cutoffs contribute mostly to the mass flux (i.e.,

a few large or many small cutoffs)? (iii) How do the transport processes change (e.g. in terms of lifetimes and frequency distributions of the cutoffs) with changing background conditions, such as a changing jet-stream or a changing thermal forcing of the ASMA?

Moreover, the correct representation of the life cycle of cutoffs from the ASMA and their coupling to chemistry is important for climate models to correctly represent the transport of anthropogenic pollution into the lower stratosphere. Therefore, the

development of detailed diagnostics for evaluating the sources, transport and chemical composition of the cutoffs, as presented in this study, is crucial.

*Code and data availability.* The source code of the cutoff analysis tool is available upon request. A public repository is planned. The CLaMS model is accessible via https://jugit.fz-juelich.de/clams/CLaMS. The CLaMS Data, as well as the analysis tools data can be obtain upon request. The ERA-Interim reanalysis is available from the ECMWF.

**Appendix A: Cutoff tracking schematic**

The tracking algorithm relates cutoffs at the two consecutive time steps $t_0$ and $t_1$. Therefore, air parcels are initialized at $t_0$ within all found cutoffs. Subsequently, the trajectories of the air parcels are calculated forward for 6 hours. The new positions of the parcels are compared with the position of cutoffs at time step $t_1$. Where there is overlap between a cutoff and the forward calculated air parcels, the cutoffs are assumed to be related. For the case of merging, forward calculated parcels of multiple

cutoffs end up in one cutoff. In this situation an arbitrary cutoff is chosen to be the parent of the merged one. Other cutoffs are ignored.





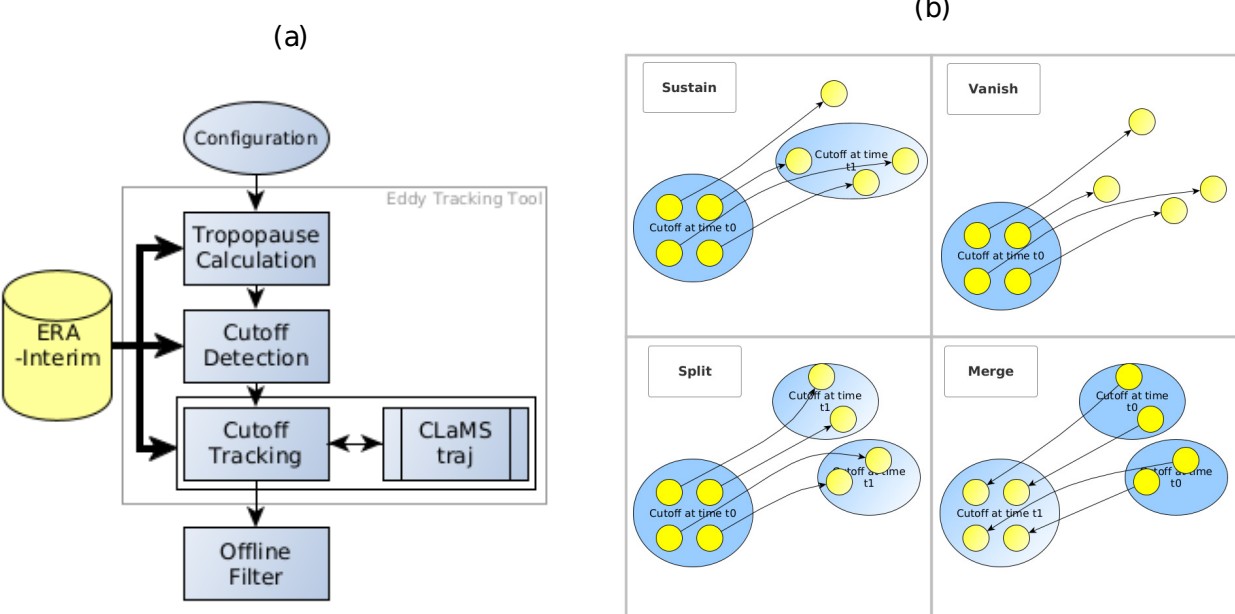

**Figure A1.** (a) Structure of the PV cutoff tracking tool and (b) a schematic of different evolutions of PV cutoffs: A cutoff can remain, vanish, split or merge. The blue circles without and with a color gradient illustrate cutoffs at a time $t_0$ and $t_1$. Yellow circles show individual air-parcels.





## Appendix B: Monthly differences in cutoff transport



**Figure B1.** Global climatological frequency distribution of ASMA related, 7.5 PVU cutoffs for each month in JJAS.

Figure B1 shows the cutoff frequency for each month of the JJAS period. In all month we found high activity over the North Pacific, with highest activity in JJA and decreased activity in September. Depending on the month, also other regions show high frequencies of cutoffs. The peak over Sibiria is strongest in June and weakens until September. It needs more investigations to estimate how much of this signal can still be related to cutoffs that are transported from Europe in eastward direction and





how much of this signal can be related to the ASMA. The most promising region and time for measurements of transported anticyclonic air masses that cross the tropopause, is the North Pacific from July to August.

**Appendix C: Validation of filter via backward calculations**

To verify the applied filter method, we initialized air parcel backward trajectories inside filtered cutoffs and analysed whether they indeed originated in the ASMA. The backward trajectories for 4.0 PVU and 6.5 PVU cutoffs were chosen to have a length of 14 days, and 30 days for 7.5 PVU cutoffs. A length of 14 days is suitable for the 4.0 PVU and 6.5 PVU cutoffs as the maximum lifetime of those was shown to be around two weeks. For 7.5 PVU cutoffs, a trajectory length of 30 days was chosen, as these cutoffs often form in the decay phase of a 6.5 PVU cutoff (see Fig. 1), and therefore may exist for up to 14 400 days longer. To define the contact with the ASMA, we set a rectangular box that corresponds to the climatological position and extent of the ASMA as similarly suggested in other studies (e.g., Garny and Randel, 2016). The box is defined to span the range between longitudes of 20° and 120°E and latitudes of 10° and 50° N. A trajectory is defined to be in contact with the ASMA if it was located within the ASMA box for at least one day (four reanalysis time steps). For each cutoff we calculated the ratio of trajectories that originated in the ASMA. If this ratio was ≥ 0.3 - at least about one third of the air came from the 405 ASMA - the cutoff was identified as a cutoff originating in the ASMA. As a result 93% of the 4.0 PVU cutoffs, 79% of the 6.5 PVU cutoffs and 96% of the 7.5 PVU cutoffs were identified as being of ASMA origin. Hence, during summer 2017 both the backward trajectory method and the empirical filter method provide very similar results, corroborating the usage of the filter method in this study.

Some limitations of this validation should be noticed. First, the exact definition of the ASMA box can slightly modify the 410 amount of trajectories that are related to the monsoon. However, as shown by Garny and Randel (2016) the simplification of a rectangular box versus a more physical PV contour for the ASMA extend does not substantially change results concerning trajectory origins. Secondly, backward calculations were limited to the 380K level, hence vertical transport was not included. On the considered time scales of a few days to weeks we deem also this approximation reasonable.

*Author contributions.* The initial conceptual idea of the study came from F. Ploeger and J. Clemens. H. Wernli, M. Sprenger and Raphael 415 Portmann helped with review and the further elaboration of key dynamical concepts. P. Konopka contributed with review and model data. The main programming work, and data analysis was done by J. Clemens. F. Ploeger supervised the study. All authors were involved in the writing of the paper.

*Competing interests.* The authors have the following competing interests: Some authors are members of the editorial board of Atmospheric Chemistry and Physics. The peer-review process was guided by an independent editor, and the authors have also no other competing interests 420 to declare.



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
