# Peer review of "Characterization of transport from the Asian summer monsoon anticyclone into the UTLS via shedding of low-potential vorticity cutoffs"

_Atmospheric Chemistry and Physics, 2021_

## Referee Comment (RC1)

The paper investigates the shedding and mixing of air masses from the Asian summer monsoon anticyclone (ASMA) with the surrounding atmospheric background at the UTLS level, using a low PV values identification method to track ASMA air masses transport.

The manuscript is very well written, with a clear logic and the figures are easy to read and well-constructed. The methods are robust and the results are of relevance in the field. I therefore strongly encourage its publication provided some minor revisions:

**General comments:** The figures are very well presented but I would warmly suggest to add some animations as well as a supplementary material. It would be very instructive to be able to follow those cutoff phases along their transport (for example for the demonstrative case studies picked for the manuscript)

Also, the idea of the schematic (Figure 1) is valuable but I found it not self-explicative. One needs to read well the whole work to understand it properly. I would suggest the following:

-having smaller arrows and shorter, to connect each different cutoff phase (while now, for example for the direct transport, the black arrow encompasses 3 cutoffs images at once, and the yellow two and there is one 7.5 PVU cutoff that is left alone)

-Adding some increasing numbers/labels to the various phases or arrows. It will make easier to follow the description of the evolution of the cutoffs.

In addition, it's not clear at the beginning of the text, that this schematic is actually summarizing the results of the study since it's presented as a first thing and there is no mention of where it comes from!

**Specific comments:**

Line 67: "For that reason"? Do you mean "For those scopes?". Also I would rephrase the sentence between lines 68 and 69 to better emphasize that you use 3 specific PV threshold each one representing a distinct feature.

Lines 71-76: I think this section here is not necessary, and it's confusing to read before the approach is properly presented.

Line 95: It has to be explained in the text why you pick the 380 K level, there is no mention about this before this section.

Line 112: TST= Troposphere-Stratosphere Transport? No previous mention of this abbreviation before!

Line 151: "..are calculated forward on an isentrope.." The 380K one?

Line 158: By "arbitrarily" do you real mean totally random choice? Is it making more sense than choosing, for example, the largest by extension or the one with lowest PV values?

Line 122: I had some troubles understanding exactly how the tracks (the black, yellow, purple lines of Figure 2,3,10) are defined. A better description of the method would be beneficial and adding the following info: are those tracks representatives of which time period (when do they start, when are they stopped)? Are they starting from the geometrical center of the cutoff?

Figure 3 caption: "the largest cutoff (3.26% of NH).." is it referred to the area? I think it's better to specify it .

Line 238: Which is the mechanism that leads to higher frequency over Siberia, is that already known?

Figure 6: It's not clear which are the units of the upper row frequencies. The axis label gives no units (which I suppose means from 0 to 1) but then the panel 6b has values up to 2 (which suggest is in %).

Line 305: has this case been chosen for some specific reason? I mean, why not using the same case as before (Fig 3)?

Line 365: "..more frequent during the early stage than during the later stages" … of what?

Lines 392-393: I think this result should be emphasized more, for example in the conclusion section. That is an important information!

Lines 405: why the 6.5 PVU cutoffs have this noticeable drop in percentage with respect to the other cutoffs? Are there other processes related to those cutoffs or are those more difficult to correctly identify?

Lines 406: The fact that the filtering was tested only for year 2017 should be specified also at the beginning of the paragraph.

**Technical comments:**

Line 239: extenT instead of extenD

Lines 248: Shown in Fig. 6a,d

Lines 253: Shown in Fig. 6b,e

Figure 2-3-10 (but also the others): The position of the described cutoffs should be better emphasized, I would suggest making the labels of the cutoff size bold, or bigger, or underlined or put a marker at their initial position. It's otherwise hard to spot them.

Figure 4 caption: "A  of 1 percent mean frequency.."

Figure 5: The dots symbolizing the starting and ending points of the track are not really well visible. Especially in the panel a, with green on green and in panel c with red on pink. I would suggest bigger dots and/or different colors

Figure 8: beware there are some typos in the legend of the cutoff PV values (panels a "cu1offs" and b "c10offs")

Line 390: Sibiria -> Siberia

---

## Author Response (AR1)

We thank both reviewers for their thorough reviews and very helpful remarks and suggestions, which helped to further improve the paper. In the following, we address and answer all of their points and explain the corresponding changes to the manuscript.

**RC1**

General Comments:
The figures are very well presented but I would warmly suggest to add some animations as well as a supplementary material. It would be very instructive to be able to follow those cutoff phases along their transport (for example for the demonstrative case studies picked for the manuscript)

Also, the idea of the schematic (Figure 1) is valuable but I found it not self-explicative. One needs to read well the whole work to understand it properly. I would suggest the following:
 -having smaller arrows and shorter, to connect each different cutoff phase (while now, for example for the direct transport, the black arrow encompasses 3 cutoffs images at once, and the yellow two and there is one 7.5 PVU cutoff that is left alone)
- Adding some increasing numbers/labels to the various phases or arrows. It will make easier to follow the description of the evolution of the cutoffs. In addition, it's not clear at the beginning of the text, that this schematic is actually summarizing the results of the study since it's presented as a first thing and there is no mention of where it comes from!

Answer:

Thanks for this suggestion. We now have added some movies to the supplement, that show the three examples of the manuscript in more detail. They should help understanding our concepts and methodology. The structure and labels of the movies is the same as for the figures in the manuscript. Within the movies we have plotted trajectories of different phases during the transport. The plotting of trajectories might differ from the ones in the manuscript, due to a change in the plotting routine (It is a bit ambiguous what cutoff of the cascade we are plotting or if we plot the geometric center of all cutoffs of a cascade for each time step, see also newly added description in the manuscript, L. 128 ff.). We've tried to design the plotting routine, such that the overall transport track can be followed easily.

Given your comment, we now have revised the schematic to make the usage of arrows more self-consistent and use numbers to refer to different pathways in the description of the schematic. We think that these changes make the schematic clearer and easier to understand.

The conceptual view of the eddy shedding process as illustrated in the schematic is now explained in L103-110, as suggested by the reviewer. It is also said there, that this separation into 2 main pathways was motivated by manual analysis of several cases and that similar transport pathways have been analysed in former studies.

Specific Comments:

Comment:
Line 67: "For that reason"? Do you mean "For those scopes?". Also I would rephrase the sentence between lines 68 and 69 to better emphasize that you use 3 specific PV threshold each one representing a distinct feature.
Answer:
We replaced "*For that reason*" with "*For that scope*" , because we agree that it is more fitting. We added: "*Each of the cutoffs represents a stage during the transport of air masses from the ASMA into the remote UTLS.*" to emphasize the different phases of the transport. (See L. 71 and L. 73)

Comment:
Lines 71-76: I think this section here is not necessary, and it's confusing to read before the approach is properly presented.
Answer:

We removed lines 71-76 to avoid redundandency and mention the filters now shortly in the next section.

Comment:
Line 95: It has to be explained in the text why you pick the 380 K level, there is no mention about this before this section.
Answer:
We now added the information that this level is representative for the ASMA in the UTLS and that the ASMA on that level can be diagnosed best from PV contours (see L. 94).

Comment:
Line 112: TST= Troposphere-Stratosphere Transport? No previous mention of this abbreviation before!
Answer:
We have written it out now.

Comment:
Line 151: "..are calculated forward on an isentrope.." The 380K one?
Answer:
We have added: … on the 380 K isentrope …

Comment:
Line 158: By "arbitrarily" do you real mean totally random choice? Is it making more sense than choosing, for example, the largest by extension or the one with lowest PV values?
Answer:
The algorithm roughly does the following: It makes a list of all cutoffs that contribute to the merged one and considers the first one in the list as the parent. The order is more ore less random. It might be affected by the fact that the algorithm starts the detection at the upper left corner. In more detail, the list is a list of indexed air parcels, such that the likelihood of a cutoff to be chosen also is increased when it has contributed to more of the indexed air parcels within the merged cutoff. We agree, that this is not the best solution, but consider it a minor issue that does not affect the main findings of our study.

Comment:
Line 122: I had some troubles understanding exactly how the tracks (the black, yellow, purple lines of Figure 2,3,10) are defined. A better description of the method would be beneficial and adding the following info: are those tracks representatives of which time period (when do they start, when are they stopped)? Are they starting from the geometrical center of the cutoff?
Answer:
We now have explained the definition of the tracks in detail (see L. 128-132).

Comment:
Figure 3 caption: "the largest cutoff (3.26% of NH).." is it referred to the area? I think it's better to specify it .
Answer:
Thanks, we now have added the specification "NH area" more frequently at the beginning to point that out.

Comments:
Line 238: Which is the mechanism that leads to higher frequency over Siberia, is that already known?
Answer:
This is an interesting question. From our analysis we see that the 4.0 and 6.5 PVU cutoffs should not play a pivotal role during this transport processes, but 7.5 PVU cutoffs should. On the other hand, our validation with backward trajectories implicates a clear relation to the ASMA, with 30 days transport times. Additionally, the jet is very strong at 60 degree east, but still sporadic cutoffs and streamers due to RWB can be created there (also visible in the study of Kunz et al. (2015), their Fig. 9).
An explanation that is compatible with this restriction is that air clockwise encircles the ASMA in its

outer area, possibly ranging far west and subsequently merges with the jet-stream on its way back. When there is Rossby wave breaking, and streamer and cutoff formation, between 30 degree and 60 degree east (see also Fig. 4 for JJA), the air mass can be transported into the stratosphere. This process is only visible from the 7.5 PVU contour, which represents the JJA tropopause.
We added a section to discuss this in detail (lines 258-263).

Comment:
Figure 6: It's not clear which are the units of the upper row frequencies. The axis label gives no units (which I suppose means from 0 to 1) but then the panel 6b has values up to 2 (which suggest is in %).
Answer:
The "frequencies" are probability densities. We have corrected the labels accordingly.

Comment:
Line 305: has this case been chosen for some specific reason? I mean, why not using the same case as before (Fig 3)?
Answer:
What makes the case of Fig. 3 valuable for our understanding, is that it is a large and long-lived 4.0 PVU cutoff. In principle we also could have used the case of Fig 2., which could have given a bit more of a coherent picture for the transitions from one stage to the next, but on the other side, since it is quite short-lived and chemical data is only available daily there would not have been a lot to learn about the chemical evolution.

Comment:
Line 365: "..more frequent during the early stage than during the later stages" ... of what?
Answer
We added: … of the overall transport process.

Comment:
Lines 392-393: I think this result should be emphasized more, for example in the conclusion section. That is an important information!
Answer:
We emphasized that now in the conclusion (L. 384ff) a bit more.

Comment:
Lines 405: why the 6.5 PVU cutoffs have this noticeable drop in percentage with respect to the other cutoffs? Are there other processes related to those cutoffs or are those more difficult to correctly identify?

Answer:
 The lower frequency for 6.5 PVU than 7.5 PVU cutoffs is likely related to the shorter trajectory calculation time of 6.5 PVU cut-offs (14 days vs. 30 days for 7.5 PVu cutoffs). This is mentioned now in the text in L. 430-432. .

Comment:
Lines 406: The fact that the filtering was tested only for year 2017 should be specified also at the beginning of the paragraph.
Answer:
We now have added it directly to the first sentence of the paragraph (L. 418ff).

Comment:
Line 239: extenT instead of extenD
Answer:
We corrected that everywhere in the manuscript.

Comments:
Lines 248: Shown in Fig. 6a,d
Lines 253: Shown in Fig. 6b,e
Answer:
We have added the correct figure labels in the lines. (See L. 269 and L. 274)

Comment:
Figure 2-3-10 (but also the others): The position of the described cutoffs should be better emphasized, I would suggest making the labels of the cutoff size bold, or bigger, or underlined or put a marker at their initial position. It's otherwise hard to spot them.
Answer:
We've added arrows to point towards the cutoffs of interest, which makes it much easier to follow the process.

Comment:
Figure 4 caption: "A frequency of 1% percent mean frequency.."
Answer: We have corrected this sentence.

Comment:
Figure 5: The dots symbolizing the starting and ending points of the track are not really well visible. Especially in the panel a, with green on green and in panel c with red on pink. I would suggest bigger dots and/or different colors
Answer:
We tried to adjust the size and color of the start and end points, but decided finally to remove them. They seem to add more confusion than clarification, and our discussion does not rely on them. To make the tracks more visible, however, we changed the color of the continents.

Comment:
Figure 8: beware there are some typos in the legend of the cutoff PV values (panels a "cu1offs" and b "c10offs")
Answer:
Thanks to point us to that. We have removed these plotting errors.

Comment:
Line 390: Sibiria -> Siberia
Answer:
We have corrected this in the whole manuscript.

**RC2:**

General comments

Comment:
Introduction: the introduction only discusses eddy shedding in general without a distinction between east- or westward shedding and the dynamical studies cited mostly discuss solely westward shedding (Hsu+Plumb, Popovic+Plumb, Amemiya+Sato, Rupp+Haynes, …). However, the analysis focuses strongly on eastward shedding events. I suggest you introduce the distinction between east-/westward shedding in more detail and cite a few dynamical studies that deal with eastward shedding.
Answer:
Thanks for this comment. We agree that the discussion of the eddy shedding process was a bit short and we tried to enhance and clarify it in the revised manuscript version. We now point out that we focus mostly on eastward shedding (see L. 42-45).

Comment:
Fig 1: the plot and corresponding discussion suggest there to be two (qualitatively) different types of eastward eddy shedding (associated with direct and indirect transport into the stratosphere). However, this

theory seems to be mostly based on the analysis of specific events like shown in Figs. 2 and 3, the rest of the results section does not really discuss this distinction. For example, I am not sure if Fig. 5 shows any indication for two distinctly different transport pathways. Do the authors think that these two pathways indeed result from two different types of underlying dynamics? As this is a main conclusion of the paper, I suggest some additional discussion/clarification.
Answer:
This is indeed a very good question. We see the distinction between these main transport pathways as a working hypothesis, that is motivated by our observational analysis, older studies (e.g. Kunz, Vogel et al.'s), and is in agreement also with the statistical results presented later in the paper. Whether the underlying dynamics are different would be a very interesting related question, but beyond the scope of this paper. To clarify these aspects, we added to the revised paper: *"The two pathways where chosen based on manual analysis of shedding events and in accordance with former transport studies (streamer: Kunz, vogel,). It remains an open question whether the underlying dynamical processes also differ."* (see L. 102-106)

Specific Comments:

Comment:
L94: what time format is used? GMT?
Answer:
We always use UTC.

Comment:
L112: TST is not explained anywhere.
Answer:
Thanks for pointing that out. TST is now explained in the revised manuscript as "Troposphere-to-Stratosphere-Transport".

Comment:
L155: how do you know that it is the "same cutoff"? I suppose your methodology actually works the other way round, so if a large fraction of trajectories initialised in cutoff X at time $t_0$ end up in cutoff Y at $t_1$ you identify cutoff Y as actually being cutoff X, right? Or is this based on something like PV overlaps?
Answer:
Yes, we are looking at the air parcels that entered cutoff Y at time $t_1$ and look from what cutoff they came from.

Comment:
Figs 2 and 3: I would suggest you include a colour bar beside the contour level description in the caption.
Answer:
We've added this certainly useful feature to the manuscript.

Comment:
L200: Maybe make clear at the beginning that this subsection focuses on all cutoffs, not just the monsoon ones, since the previous subsection just discussed how you identify monsoon cutoffs.
Answer:
To clarify this, we added it now to the first sentence of the chapter (L. 212-214)

Comment:
L224-228: you need to be careful here with your phrasing because Fig 5 does only show the frequency of cutoff occurrence, not actually the transport. Also, do you have an idea why you only find a weak signature of westward shedding (because other studies (eg Popovic+plumb) rather suggest a dominance of westward over eastward shedding? Could this relate to the details of your identification algorithm? Or does it have something to do with the zonal extent of the climatological monsoon PV low already including a signature of westward shedding (eg the region <60N) and hence such signatures are hard to identify?
Answer:

We agree with the reviewer's reasoning why we might not distinguish westward shedding from the ASMA climatology and added a sentences to mention that possibly the signal of westward shedding is hidden and undersestimated, respectively (see L. 240 ff.).

Addtionally, we have rephrased the parapgraph a little bit, to emphasize more that we use the distributions as an indication for the transport pathways, but not show the actual transport.

Comment:
L228: Hsu+Plumb only study westward shedding, so they do not say anything about the eastward shedding signatures.
Answer:
We are now refering to Rupp&Heynes 2021, who deal with east- and westward shedding. Addtionally, we also have distinguished more precisely between eastward and westward transport, which also adresses one of your previous comments. Hsu+PLumb now is only refered to for the westward shedding. See Lines 233-242.

Comment:
L239: should say "… larger extent of the former cutoffs."
Answer:
Yes, we changed that.

Comments:
Fig 5: you should either extend your colour bar to include a shading for >5% or indicate that these values are coloured white. Also, the cutoff tracks are quite hard to see with the coastlines also being thin black lines.
Answer:
Fig. 5 : To make it clear that values above 5% are marked white, we have indicated that in the caption. To make the tracks more visible, we have colored the continents gray. Furthermore we removed the green and red dots, because we believe, it produced more confusion than clarification. We, however, changed the color of the coastlines to make the black lines more visible.

Comment:
Fig 6: the grid in the lower row plots should probably be moved to the background, otherwise it looks like the lines are discontinuous. Also, is there a reason why panels c and f use a different line width?
Answer:
Taking into account your comments, the plots are looking better now, without "grid problems" and with the same linewidth.

Comments:
Fig 8: The legends should be checked again. And maybe combine the two separate entries for "entire atmosphere", otherwise this might be confusing.
Answer:
Fig. 8 : We removed one of the labels "entire atmosphere", as we agree that two labels are not necessary. The errors in the labels have been removed.

Comments:
Fig 9: is it reasonable to include some sort of reference lines here, eg values typical for the bulk monsoon anticyclone and for the troposphere/stratosphere, so the reader can see a gradual convergence from one to the other?
Answer:
Fig. 9 : We considered to insert some reference lines but then the plots became too busy and we left them unchanged.